# Grief Iconography between Italians and Americans: A Comparative Study on How Mourning Is Visually Expressed on Social Media

**DOI:** 10.3390/bs11070104

**Published:** 2021-07-20

**Authors:** Illene Noppe Cupit, Paolo Sapelli, Ines Testoni

**Affiliations:** 1Department of Psychology, University of Wisconsin—Green Bay, Green Bay, WI 54311, USA; cupiti@uwgb.edu; 2Department of Philosophy, Sociology, Pedagogy and Applied Psychology (FISPPA), University of Padua, 35131 Padova, Italy; paolo.sapelli@studenti.unipd.it

**Keywords:** death, grief, internet, photograph, comparative study, social network sites

## Abstract

As an innovative way to express grief, social media posts about the deceased have become fairly common. However, few studies have examined commonly posted grief photos. The purpose of the present study was to examine such pictures, as well as the motivations and reactions of those who posted them, among Italians and Americans. Surveys were sent to both Italian and US participants. The US group yielded 262 responses (mean age = 22 years; 81% female), and the Italian group yielded 51 (mean age = 32 years; 82% female). Several key issues emerged, such as the need for social media users to receive empathic support from other users, the desire to maintain continuing bonds, the wish to remember the deceased, and the desire to share beauty and symbolic pictures. The images were analyzed using content analysis. Both samples posted photos to remember, and to enhance their posts. A strong preference for pictures with a positive emotional connotation appeared, depicting the deceased in a conjoint appearance with the participant. The results suggest that the imagery used for the expression of grief in social media sites, an “iconography of grief”, is a popular means of expression for grievers across the two cultures.

## 1. Introduction

### 1.1. State of the Art

The expression of online grief is becoming increasingly common [1]. The first Web memorial (“Web cemetery”) was opened in 1995 [2], and it developed a continuity between emotions, reminiscences and the establishment of communities to commemorate the deceased [3,4]. These digital cemeteries were joined by sites, blogs and forums for the manifestation and processing of mourning in public spaces. In the 21st century, the birth of social network sites (SNSs) [5] enabled grief to become more communal, and brought death back into everyday life [6]. The collective funeral celebrations of public personalities [7] and the individual commemoration of deceased loved ones in specific support online groups [8,9,10] have increased considerably, and have been extensively studied. Looking at Facebook in particular, it is possible to ascertain how the two-way communication typical of new media has allowed anyone to participate in someone’s grief, normalizing what would once have been regarded as a rude invasion of immediate family mourning [11]. Furthermore, according to Brubaker and colleagues [12], social media have expanded mourning in the course of digitalization due to the combination of user’s networked communication and SNSs’ automated notifications. In particular, they identified three main areas where online mourning has expanded: the temporal, enabled by the asynchronous nature of SNSs and resulting in both the immediacy of information enabled by the daily use of SNSs and the breadth of information available as individuals add content from the past and present (e.g., discovering the death of friends and contributing postmortem comments); the spatial, enabled by the removal of geographical barriers, allowing distant users to interact around death and bereavement; and the social, referring to the dissemination of information across previously separate social groups unified by SNSs. Finally, a fourth form of mourning expansion has been identified: cultural expansion [13]. This form is closely linked to the social, and is based on the negotiation and appropriation of mourning’s norms within social media, leading to an expansion in the array of rituals and practices related to death even in the offline world.

Before the internet era, ceremonies such as funerals provided family members with specific spaces of commemoration with a limited time-frame, but in the “social media age” SNSs provide a public venue with a potentially much wider audience for commemoration, co-constructing biographies of the deceased and fostering a continuation of the relationship with the deceased [14,15,16,17]. Indeed, the death of a significant other is not the end of the relationship; rather, the relationship persists, not frozen in time, but evolving with modifications of biographies of the mourners and of the deceased [18]. Such a continuity in the relationship is facilitated by the media, which can go beyond time and space, and ultimately beyond life and death. In fact, due to the mass of today’s information, SNSs play a fundamental role in defining how we understand death and how we face it, presenting themselves as the tools with which new death-related rituals gain popularity. The expression of mourning via an SNS provides a manifestation of one’s feelings of grief, empathy and condolence in new multimedia ways that are no longer limited to speech or to text on paper. Furthermore, because the digital dimension does not force people into ‘face-to-face’ interactions, it removes many of the main interpersonal risks that accompany this type of communication. It follows that suffering a loss online is usually “safer” from an interpersonal point of view, and that it allows not only family but also friends and acquaintances [3,19] to discuss the deceased’s life with less inhibition and to choose what kind of words to use to avoid embarrassment or emotional tension [15]. The connection between grief and the internet is becoming so strong that some authors have begun to speak about “social media mourners” to refer to those people who, having lost someone, make use of social networks to face the loss through one-way communication (to express mourning), two-way communication (to dialogue on death with others) or immortal communication (to communicate with the deceased him/herself) [20]. However, because in this article we refer to (semi-)publicly displayed grief posts, we rather prefer to speak about “multi-way communication” in order to highlight the interactive nature of social media and the participation of several users in the same communications.

In Western societies, traditional forms of death ritualization [21] have become increasingly removed from their religious roots, as well as being truncated in time and accessibility to family members who may have moved far from their families of origin [22]. Communal support by physical presence has been particularly affected by the COVID-19 crisis, which rendered grievers socially isolated [23]. The process of a deceased’s biographical reconstruction may result in a number of roadblocks [24] that SNSs seem to solve. The digital landscape offers valuable solutions through a greater connection between people, as well as new traditions within the framework of traditional and non-traditional religions [7,25,26,27,28,29]. SNSs may serve to reintroduce death into the world of the living by allowing the sharing of stories with others in order to cope with the loss [30,31].

The majority of the research in this field has focused mainly on the verbal content observed or reported by the bereaved. However, SNSs offer the possibility of the visual manifestation of grief by publishing photographs or images related to the deceased, as photography is one of the simplest ways to remember [32,33,34]. From a historical perspective, the use of postmortem photography, also called memento mori photography, was widespread in the United States during the 19th century [35]. Contemporary practices, particularly with regard to parents posting pictures of their deceased children, are an online manifestation of this earlier behavior [10]. The research on photographs in relation to online mourning has spanned the gamut from funeral selfies, which may communicate an individual’s affect to a broad audience [36], to postings which include photos of loved ones who died by suicide [37]. Church [38] writes of the “digital gravescape” on Facebook, where photographs show visual depictions of the afterlife and nature scenes. Images appear to supplement the poems, song lyrics and personal updates from adolescents mourning their peers on an SNS [38]. Additional studies published on this “iconography of grief” have indicated that the photographs are useful for coping with grief over missing persons presumed dead, and with the intense grief over deceased children [39,40]. Most of these studies found that the verbal and visual postings were used to communicate with the deceased, as a “continuing bond”. Overall, grief photography is considered an ‘‘evolving practice”, both in the online and offline world.

Based on the literature which makes use of visual material as a valid source of information [3,10,41,42,43,44], an analysis of the visual content for coping with the death of another that was shared online—comparing Italian and US users of social media—became the focus of the present study. In particular, the performance of a virtual ethnographic study using a content analysis [41] of grief iconography expressed in social media sites in two culturally different scenarios could, compared to content analysis of written expressions of grief, further the understanding of how grief is expressed in online public spaces. Additionally, the present research has strong implications in the current pandemic state, in which Internet resources are the ‘only’ possible ways of communication for many grievers.

### 1.2. Aims of the Study

The study focused on Italian and US mourners’ reasons for posting pictures of their deceased loved ones on social media. We also considered differences in the content of the images published, and in the rationales behind such postings. The possible differences between family and religious representations were also considered. The research followed the APA Ethical Principles of Psychologists and Code of Conduct [45]; the participants received a detailed explanation of all of the objectives of the research and the methodology of analysis used. They were asked for permission to use their data, to transcribe their answers and to analyze their content. The anonymization of the content of the obtained texts was assured, and only those who gave online or written and signed consent participated in the research. The study was approved by the Ethics Committee for Experimentation of the University of Padua (n. F3AF42BC04992E7B78CE369D49BAF14D), and by the Institutional Review Board of the University of Wisconsin-Green Bay.

## 2. Materials and Methods

### 2.1. The Research

The study belongs to the area of qualitative research in psychology and the social sciences. Upon consultation with US colleagues and students familiar with the online postings of deceased individuals, a survey was created that questioned participants’ motives for posting an online picture of a deceased loved one. In addition, the participants were asked to upload the picture in a space provided in the survey. While the pictures were analyzed via content analysis based on categories derived from the research questions (top-down approach) [46], the verbal responses to the survey were analyzed via thematic analysis, wherein codes were developed directly from the data (bottom-up approach). This procedure requires the researcher to become familiar with the data through multiple readings, so as to achieve a detailed, rich and complex reorganization and description of the responses [47,48].

Overall, the research used a mixed-method design, which is growing in the area of psychological studies. This method combines quantitative and qualitative patterns to analyze narrations, concepts and representations. Despite the arguments that qualitative and quantitative methodologies are polarized and cannot be mixed because of different epistemological perspectives [49], as indicated by de Block and Vis [50], the transformation of qualitative into quantitative data can be useful, especially for qualitative comparative analysis (QCA). QCA is a practical synthesis of qualitative data through the translation, transformation, or conversion of qualitative data into numerical data, which permits it to have the minimum information sufficient and necessary to overcome the incommensurability between different fields of sense and meaning [50]. The adoption of the QCA required the agreement of the work among researchers to determine the thresholds for inclusion and exclusion of the sets of elements in the list of categories recognized.

### 2.2. Instrument and Participants

A survey of 25 questions (in the English and Italian languages), including 2 questions on gender and age, 8 open-ended questions, 14 closed-ended questions and 1 question asking the participant to submit a photo previously posted on an SNS, was shared in Italy and the United States on Facebook. Created in English, translated into Italian and then back-translated in English, the survey (implemented via the Qualtrics platform) investigated participants’ intentions, expectations and reactions about posting grief content. Questions about the nature of the participants’ relationship with the deceased were also included. The participants were explicitly invited to submit and answer questions about the photos they posted on Facebook, Twitter and/or Instagram.

The data from the US and Italian samples are presented in Table 1. Thirty-seven percent of the Italian participants agreed to submit an image of a deceased loved one published on social media, for a total of 19 images. For the US sample, 21% provided an image, for a total of 54 pictures.

All of the statistical analyses were performed with the IBM SPSS Statistics software package.

### 2.3. Method

#### 2.3.1. The Analysis of the Open-Ended Questions

A thematic analysis was conducted on the responses, which were then grouped into several themes [38]. The themes were derived from both the Italian and US data, and discussed by the researchers from both cultures, until a final, single set of themes was developed. The themes captured the participants’ reported experiences, meanings and/or descriptions, taking into account the research questions regarding bereavement and mourning. Before the coding of the entire data set, the reliability of the themes was ascertained via independent codings between two of the authors on the responses of 14 randomly selected participants. Based on the total number of matches and non-matches, the final coding reliability equaled 68%. Considering that, in most of the non-matches, the two raters coded the same way and differed only in the number of codes used, the interrater reliability was considered to be acceptable for further analysis.

#### 2.3.2. The Analysis of the Images

The photographs were subjected to content analysis based on a top-down approach driven by the research questions. We developed thirteen analytical categories focusing on the content and stylistic structure of each image. Furthermore, the respondents were asked to describe the image in words. Together with the image, the verbiage was used as additional support to better understand what was depicted in the image, and to apply analytical categories. In order to maintain the cultural integrity, different authors analyzed the images of their respective cultures.

## 3. Results

Table 2 presents the most used SNS for bereavement purposes, the deceased most commemorated online, and the comparison between the actual age at death and the age of the deceased in the shared image.

The thematic analysis on the texts obtained from the participants’ answers resulted in a total of 73 themes (see Table 3 for the dominant themes that emerged from each question). Table 3 also describes the operational definition of each theme and their percentages for the Italian and US samples. Some of the participants’ responses did not lead to the development of any theme due to the idiosyncratic nature of their comments.

The remaining themes were grouped into five prevalent thematic areas that are described below together with the results of the pictures’ content analysis.

### 3.1. First Thematic Area: The Reasons for Showing or Not Showing the Photos

The first thematic area is inherent to the reason for showing or not showing photos. When the participants shared a bereavement post without a photo, it was because they wanted “to feel part of the community” (It 11%; US 12%), calling for empathy and humanity from those who are part of virtual community, and eventually to receive and give support to those who are grieving. Other reasons given were “to think about death” and to express beliefs regarding death, mourning and/or the deceased (It 17%; US 0%). The participants also desired “to inform everyone” about the deceased’s identity and their death, the funeral rite schedule and other information regarding that person’s passing (It 6%; US 11%). Some of the respondents wanted “to express emotion” related to the loss (It 17%; US 16%). Adele noted:
*do not believe a photo is necessary to post during the down times. I made posts and comments without photos to use words to express my grief.*

When the participants posted a photo, some mentioned that it was because they wanted to commemorate the deceased on anniversaries and “to remember” (It 16%; US 12%). In other cases, it was due to the belief that simply writing a post wasn’t enough, and so the picture helped “to convey the essential” (It 17%; US 4%). Other times, the participants wanted to call for empathy from users that are part of the virtual community, to give support to other grievers, and so to “feel part of the community” (It 0%; US 1%). The participants also wanted “to inform everyone” (It 0%; US 7%) or “to express emotion” related to mourning (It 8%; US 5%). For example, Julia expressed her desire to remember her deceased loved one:
*After I write my feelings down and describe everything good about him I would post a picture of him to make sure everyone who read will remember him as a good person so he won’t die in our memories.*

Several participants never posted a photo to cope with death or grief whilst believing that was the way “to convey the essential” (It 7%; US 17%) or because they were “unfamiliar with the SNS” (It 7%; US 9%). Others never posted a picture because they judged such practices to be “inappropriate” and exploiting a tragic event (It 13%; US 11%). The choice to publish the photograph on a particular day was intended to “receive support and lessen the pain” (It 8%; US 6%); “to express the void has been left” (It 6%; US 6%), conveying nostalgia and melancholy; or “to bequeath” (It 4%; US 0%) teachings, life experiences and quotes that were learned from the deceased, or from their passing. Other times, the choice was dictated by a “coincidence” (It 2%; US 2%), in so far as participants associated an event with the deceased, or by the desire to “to think about death” (It 13%; US 4%), expressing thoughts or beliefs regarding death and/or the deceased. Finally, a little evidence of “continuing bonds” (It 4%; US 1%) was highlighted by the participants who wanted to continue the relationship with the deceased by sharing a picture on that particular day.

### 3.2. Second Thematic Area: The Main Comments on the Photographs

The second thematic area refers to what the participants wrote on the SNS about the photo they shared. The participants’ responses demonstrated evidence of “continuing bonds” (It 11%; 15%). Additionally, the participants wrote comments to “refer to eternity” (It 10%; US 1%) and to a timeless dimension where the deceased will never be forgotten. Others used the SNS “to think about death” (It 7%; US 15%) or “to quote something” (It 10%; US 3%), such as a song, a poem, or a movie. Some participants addressed the deceased with an “intimate nickname” (It 8%; US 2%) that defined the kind of relationship they once had (e.g., “my beloved”, “mentor”, “angel”); they also “expressed disbelief” in accepting the reality of the death (It 3%; US 0%), or “regret” (It 3%; US 2%) over missed opportunities with the deceased and/or how the death occurred. The participants expressed their felt pain over the death as “the void which has been left” (It 7%; US 3%). A few tried “to make the community aware” (It 2%; US 1%) by inviting users of the virtual community to act in a specific way and/or to be empathetic. Maria offered a typical example, wherein she said: “I wrote the whole St. Augustine’s poem ‘*death is nothing’*.”

### 3.3. Third Thematic Area: The Choice of Photography

The third thematic area refers to the reasons behind the choice to share specific photos online. The participants chose photos they found to be particularly important, representative of the deceased and suitable for an “important statement” (It 15%; US 7%). Enrico stated:
*Because it is a very representative photo of her, she is smiling and looks like she is dancing on the beach, which she particularly loved to do.*

Furthermore, the participants chose photos depicting “special occasions” (It 7%; US 4%), such as a party or celebration related to the deceased (e.g., a marriage, a graduation day), or photos of the “first picture (or one of the first)” (It 2%; US 1%) or the “last picture (or one of the last)” (It 9%; US 5%) of the deceased. The participants also chose photos because of their specific characteristics; because of their depiction of an unknown, unique side of the deceased (It 11; US 13%); or because of their depiction of both the participant and the deceased in a “conjoint appearance” (It 9%; US 21%). Surprisingly, there was little evidence of “continuing bonds” (It 2%; US 1%).

### 3.4. Fourth Thematic Area: Comments Expected and Actually Received

The fourth thematic area refers to the comments which the participants anticipated and actually received from other SNS users. This thematic area also refers to the participants’ thoughts and feelings about the received reactions. The comments expected by the participants were related to the hope to be welcomed and to receive “appreciations” (It 2%; US 3%), and to gather memories and thoughts of the deceased or “commemorations” (It 16%; US 16%). Comments about the “emotional experiences of others” (It 16%; US 11%) were seen as well. The participants hoped to receive “condolences” (It 10%; US 14%) and wanted to maintain a relationship with the deceased through a “continuing bond” (It 5%; US 2%). For example, Patricia reported her hope to gather commemorations of her loved one:
*I wanted my family to be able to react to the photo and remember him in a positive way. Also, I wanted family to share comments about their good memories.*

The comments and reactions actually received were very heterogeneous: many participants recieved “likes” (It 23%; US 18%), “heart reactions” (It 14%; US 8%), or comments welcoming and “appreciating” the picture shared (It 2%; US 6%). Other comments received pertained to memories paying homage to and “commemorating” the deceased (It 11%; US 9%), or to the “emotional experiences of others” (It 19%; US 18%) relative to someone passing. A few comments received were judged by the participants as “inappropriate reactions” (It 2%; US 2%) because they were rude, offensive or inauthentic, as in the case of Eliza:
*People he wasn’t even close with started commenting like ‘I’m gonna miss you so much’ and ‘you were like a brother to me’ when in reality they barely talked.*

The participants’ thoughts and feelings about the received reactions involved not only “positive outcome”, “negative outcome” or “human warmth” (see Table 3) but also “moral considerations” (It 5%; US 2%) judging comments as right or wrong, or “appreciations for the commemoration” (It 13%; US 9%). The latter case was positively received because the memory of the deceased was maintained and/or collectively reviewed.

### 3.5. Fifth Thematic Area: The Subject of the Shared Images

The fifth thematic area focused on the analysis of bereavement pictures shared on SNSs through the use of 13 analytical categories. All of the bereavement images were distinguished as photographs, and were thus labeled “picture” (It 96%; US 93%), or as a drawing or illustration, and were thus labeled as “image” (It 4%; US 7%). The percentages of the other 11 categories were calculated separately because they specifically related to shared representations of “persons” (It 27%; US 35%), “objects” (It 4%; US 4%) or “settings” (It 6%; US 6%). Some cases were coded as “refined” (It 6%; US 1%) as a result of modifications made with the use of programs (e.g., cutouts, filters, overlaid images, added text, or special effects). Shared images frequently depicted the participants in their “early youth” (It 4%; US 3%), people in “acts of kindness and care” (It 10%; US 4%) such as kissing and hugging each other, staring into each other’s eyes, or acting in a nurturing manner. Other photos showed the participant and the deceased together in a “conjoint appearance” (It 16%; US 19%). The last three categories referred to the emotional connotation (when clearly understandable) of the shared images: “positive” (It 15%; US 17%), “neutral” (It 3%; US 1%) and “negative” (It 0%; US 0.5%). In the Appendix A there are some sample images that participants sent to us and authorized us to use for research purposes. For both samples, close-up pictures of the participant’s hand intertwined with that of the deceased were frequently seen. Online mourners may have used such pictures as a symbolic display of affection for the deceased.

## 4. Discussion

Social media sites such as Facebook have many pictorial postings that reflect grief and loss. Frequently, these are accompanied by verbal content by the author and the viewers of the photos. Such postings may represent a new norm in the mourning process that simultaneously adheres to traditional practices of commemorating the dead via pictorial representations, as well as to the reconstruction of these norms on a virtual platform [13]. However, few studies have been conducted that include this iconography associated with postings about death. The present study makes a threefold contribution to understandings of the iconography of grief by: (1) examining pictures and the verbal content of grief-related postings on Facebook, Instagram and Twitter; (2) querying the authors about their motivations behind the selection of their pictures; and (3) examining the differences and similarities between Italians and Americans regarding this online expression of mourning.

Starting from the first thematic area (reasons for showing or not showing pictures), both samples showed a strong tendency not to share anything concerning a deceased loved one in order to protect their privacy, because this practice was perceived as intrusive and excessively public [42,44,51]. Furthermore, Americans seemed to make greater use of the communicative potential of social media and the “collapse of the context” [52] to report deaths [20]; in fact, Americans reported higher percentages in relation to the willingness to inform everyone about the death, and in relation to the choice of images that adequately portray the deceased’s appearances to a vast and varied virtual audience. For those participants who did post on SNS, the evidence spoke to the tendency, typical of mourning posts, to speak to the deceased instead of about the deceased [12] (the second thematic area). Many Italian participants, in fact, addressed the deceased directly by expressing a greeting, a wish, a dedication or a promise to the deceased (coded as “to pay their respects”). Similarly to the US participants, the Italians used SNS communication mostly to express emotions and to remember. Some participants from both samples addressed the deceased love ones with intimate nicknames (coded as “intimate nickname”), referring to the deceased as an “angel” in 50% of these cases [28,29].

The findings in the literature attest that the main motivations to share negative emotions are to support one’s coping skills and to reduce one’s emotional load [53]; the reported choices to publish the photo on a particular day seemed to correspond to the second motivation rather than the first. The findings of the third thematic area (choice of photography) corroborated the results of Keskinen and collaborators [10] by indicating that the photographs were posted to preserve the emotional–relational bond with the deceased (coded as “continuing bond”). The majority of the participants in both samples had “no expectations” for the responses they anticipated receiving about their postings. Perhaps this signifies that the mourners used the SNSs as a free space to express their emotions in a cathartic way without expecting anything in return. They did, however, receive high rates of empathic support [20].

The codes emerging from the reactions that the participants actually received, such as “empathic support” and “emotional experiences of others”, reflected the exchange of hope found to be typical of users engaged in grief online [54]. Furthermore, both samples scored high percentages for the “likes” received, and very low percentages for “inappropriate reactions”. These participants, therefore, benefitted from addressing the issue of death on the web, a finding supported by the research of De Vries and Rutherford [19]. The participants’ thoughts and feelings (the fourth thematic area) about the received reactions led to important insights about the perceptions and emotions surrounding sharing posts related to deceased loved ones. The consideration of the low percentage of “negative outcome” together with the high percentages of “positive outcome”, “human warmth”, and “appreciations for the commemoration” suggests that bereavement posts are, in most cases, well received by virtual bystanders. Both samples benefited from the communication of mourning via SNS, but the Americans derived greater benefits and suffered fewer negative consequences from sharing than did the Italians.

It should be noted that although the use of Twitter is increasing in Italy, it seems that the use of Italian users of this social medium does not extend to the manifestation of mourning and the celebration of the dead. The primacy of this practice, unquestionably, was Facebook dominant. The disparity between the age at death and the inferred age of the deceased in the photographs that were shared online seemed to highlight a certain “rejuvenating effect” of the deceased. Both the US and Italian samples tended to portray the dead as younger. However, further studies in the future will need to investigate this rejuvenating effect of photographs of the deceased that was observed on SNSs. Italy is experiencing a phenomenon known as “famiglialunga” or “long family” [55,56], characterized by the tendency of young adults to live at home with their parents between the ages of 18 and 35, or even later. This demographic trend may have implications for the manifestation of grief online. When the deceased was a parent (mother or father), the coding percentage of the category “conjoint appearance” was 54% for the Italian sample and 0% for the US sample. Regarding grandparents and great-grandparents, the percentage was equal to 60% and 58%, respectively, in the Italian and US sample, and as for friends (male, female or other) the percentage was 43% in the Italian and 60% in US one. Finally, the second thematic area also suggested that Italian people attested higher percentages in themes which can be considered religious or spiritual, such as “refer to eternity”, “expressed disbelief”, “regretted” and “the void which has been left”.

## 5. Conclusions

Pictures and visual symbolic imagery have evolved from 10,000 year-old petroglyphs found in Central Asia to the current globally used electronic forms of communication seen in social media sites, texting and websites. The examination of the content and motivations behind mourner’s postings contributes further information to the growing body of literature on the iconography of grief [36,37,38].

Stroebe and Schut [57] proposed a dual process model of grief wherein mourners oscillate between emotion-focused coping (that which was lost) and restoration-focused coping (adapted to a new life without the physical presence of the loved one). Both of these processes were evident in how Italians and those in the US relied on social media to post pictures of the deceased. In this new iconography of grief, the photographs and images often were accompanied by emotional language, and the postings themselves served a restorative function in their attempt to learn to live in a new assumptive world.

For both the Italian and the US sample, Facebook was the most-used SNS for mourning expression. The Italians paid more homage to parents on SNSs than did the Americans, but in both cultures the deceased most commemorated was the grandmother. In the Italian sample, the choice to publish an emblematic image of a certain experience was most evident, whereas the US sample showed a greater preference for a beautiful and harmonious representation of the deceased. In both the Italian and the US samples, most of the participants expressed few expectations from others regarding their posts relating to their deceased loved ones. Overall, there was a remarkable similarity between the Italians and Americans: they both preferred to share emotionally positive pictures depicting the deceased, whilst occasionally appearing in the pictures as well.

The findings of this study were limited by the selection of convenience samples from both cultures; the participants were not necessarily representative of their relative reference countries. Furthermore, we cannot know if the observed differences were really due to nationalities or other external factors such as occupations, religions or socioeconomic backgrounds, because we did not control those variables for the Italians or Americans.

This study attempted to further the understanding in the research literature on the photos and images in Facebook postings of the deceased; the research used a grounded theory approach [20] (inductive content analysis) as opposed to hypothesis testing. Certainly, further research should be conducted in order to validate the codes discerned from the present data, and to further test the hypotheses drawn from dual-process theory. Comparing two cultures on social media and grief presented a unique opportunity to study online grief from a global perspective. It is hoped that this research inspires others to look at the ways in which SNSs are used in different countries to cope with grief and mourning. Finally, the current study suggests that follow-up research should pursue the ways in which the religiosity and spirituality of the participants play into the iconography of online grief postings, particularly with regard to beliefs about the after-life [58], coping strategies, empathy, moral judgments and religious proselytism [7].

## Figures and Tables

**Table 1 behavsci-11-00104-t001:** Participants.

Samples	Total	Female	Male	Other	Mean Age	Standard Deviation
US	262	81%	17%	2%	22.09	6.2
Italy	51	82%	16%	2%	32.178	12.439

**Table 2 behavsci-11-00104-t002:** Percentages of different uses of SNS in the Italian and US groups.

		Italian Group		US Group	
		Frequency	Percentage	Frequency	Percentage
SNS used for bereavement purposes	Facebook	42	74	150	51
	Instagram	15	26	115	39
	Twitter	0	0	32	11
Kind of deceased honored via SNS	Grandmother	10	22	13	23
	Grandfather	3	7	9	16
	Mother	7	15	1	2
	Father	6	13	3	5
	Friend (M)	5	11	12	21
	Friend (F)	3	7	0	0
	Other	13	25	19	33
Made use of more than one SNS for bereavement purposes		11	19	82	28
Published a picture of deceased on more than one SNS		12	22	33	22
Actual age at time of death	Age range (years)				
	0–20	2	5	20	19
	21–30	4	10	5	5
	31–40	1	2	4	4
	41–50	5	12	7	7
	51–60	10	24	10	9
	61–70	5	12	12	11
	71–80	7	17	22	21
	81+	7	17	26	25
Age of deceased in the shared image	Age range (years)				
	0–20	4	14	16	20
	21–30	3	10	4	5
	31–40	3	10	8	10
	41–50	4	14	6	8
	51–60	4	14	12	15
	61–70	7	24	8	10
	71–80	2	7	15	19
	81+	2	7	11	14

**Table 3 behavsci-11-00104-t003:** Percentage of the main codes found in response to the open ended questions.

Question	(Y/N)	Developed Theme	Operational Definition	Percentage in Italian Sample	Percentage in US Sample
Have you ever posted to cope with death, dying, and grief without a photo? Why or why not?	YES	To remember	The participant found commemorating and honoring the person who passed away on anniversaries by sharing a post without a picture.	11	30
		To convey the essential	The participant shared a post without a picture as they found a picture to be excessive, unnecessary, morally wrong or generally violating his/her own privacy or the privacy of the deceased.	33	26
	NO	To avoid it being made public	The participant did not share any kind of post regarding the deceased, death, dying or mourning as they considered doing so morally wrong, meaningless, harmful and/or inappropriate for intimacy and privacy.	42	66
If you have never posted a photo on Facebook, Instagram, or Twitter to help cope with death, dying, or grief, why not?		Because it is considered futile	The participant did not share a picture to deal with death, dying and mourning as they considered doing so useless, unnecessary and unsatisfying.	40	28
		In the interest of intimacy	The participant did not share a picture to deal with death, dying and mourning as they considered it an invasion of their privacy and intimacy and/or because it was a topic which they consider too personal to discuss publicly.	33	34
Did you write anything about the photo?		Expressing emotions	The participant recounted the emotions provoked by their loss to vent and/or express frustration, sorrow and the fact that they missed the deceased. The participant may also recount positive emotions.	13	36
		To remember	The participant recounted one or more memories that have to do with the deceased, the person, what they have done and overall their place in society. They did so in order to honour and commemorate the person (this was why the participant often shared this kind of content on anniversaries).	13	18
		To pay their respects	The participant sent their regards, their best wishes, an inscription or a promise to the deceased.	13	5
What were your reasons for posting this photo on that particular day?		To remember	The participant shared the picture on that specific day to recount one or more memory which has to do with the deceased, the person and overall their place in society. They did so in order to honor and commemorate the person. This was why the participant often shares this kind of content on anniversaries.	37	47
		To express emotion	The participant shared the picture on that specific day to freely express their pain, sorrow, anger, frustration, the fact that they missed the deceased or any other emotion.	10	19
		To inform	The participant shared the picture on that specific day in order to inform everyone about the deceased’s identity and their death, how it occurred, the funeral rites schedule, society’s reaction and other information regarding that person’s passing.	13	16
Out of all the photos that you could have chosen, why did you pick this particular photo?		Beauty of the picture	The participant had chosen to share this specific picture as they found it adequately portrays the deceased’s appearance (good-looking, photogenic, nice smile…) or because it was considered a nice picture.	15	25
		Emblematic picture	The participant had chosen to share this specific picture as they found it to be emblematic and strongly representative of a certain experience, emotion, cognition or moment.	28	21
What reactions or commentswere you hoping to receive by posting this photo?		No expectations	The participant had no expectations concerning reactions or comments referring to the picture.	30	32
		Empathic support	The participant expected to receive reactions and comments of a supportive and helpful nature and/or empathy, humanity or comprehension.	23	22
What reactions or comments did you actually receive regarding posting this photo?		Condolences	The participant had received condolences.	7	21
		Empathic support	The participant had received reactions and comments of a supportive and helpful nature and/or empathy, humanity or comprehension.	23	18
What were your thoughts and feelings about the reactions that you received (e.g., likes, comments) from posting this photo?		Positive outcome	The participant experienced positive thoughts and feelings that could lead to wellbeing, satisfaction, appreciation and enjoyment and was overall satisfied with the feedback.	33	43
		Negative outcome	The participant experienced negative thoughts and feelings that could lead to uneasiness, disappointment and melancholy and was overall unsatisfied by the lack of support.	13	7
		Human warmth	The participant experienced thoughts and feelings that made them feel emphatically supported, understood, loved and not alone when dealing with pain.	20	30

## Data Availability

The datasets used and/or analyzed during the current study are available from the corresponding author on reasonable request.

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
