# Peer review of "Grief Iconography between Italians and Americans: A Comparative Study on How Mourning Is Visually Expressed on Social Media"

_behavsci, 2021, doi:10.3390/bs11070104_

Round 1
Reviewer 1 Report
This study addresses ongoing changes in people's behaviors during the losses of loved ones in new socio-technical landscapes. While a majority of related studies have focused on text data, this study analyzes how image sharing has occurred and why. I found this article interesting and important since it can pose various future study opportunities.
However, I believe the authors needed to address the following revisions for publications.
- Misspelling, unfinished sentences, styles
- Line 15: please complete the bracket with the percentage of females.
- Line 16. “The need” for whom? Social media users? Researchers?
- Line 68: [33]. [26] should be [33, 26]
- Line 78: (31, 32 ) should be [31, 32], “to create” —> “To create”
- Clarification of the study's contribution
- Line 21: 'Results suggest that imagery used for the expression of grief in social media sites, an “iconography of grief,” is a popular means of expression for grievers. ' --> These results sound not surprising or pose new discussion. Please consider describing what are the takeaways of this study for future studies compare to previous works.
- Some statistical analyses are needed for discussion. For example, the authors discussed the tendencies between nationalities (Line 291-292: “American seemed to make …” ) but no statistical analyses are conducted
- I believe that the authors need to carefully discuss the results in terms of participants' socio-economic backgrounds. The authors mentioned the differences between participants from Italy and the U.S. However, are the differences really due to nationalities? Are there any possibilities of differences in participants' demographic and socio-economic backgrounds? For example, I was wondering whether other external facts, such as gender, ages, occupations, religions are well controlled over the participants.
Author Response
As you recommended, we have applied your proposed revisions (see the document linked to this form)

Reviewer 2 Report
The authors perform a virtual ethnographic study using content analysis of grief iconography expressed in social media sites in two culturally different European and American scenarios. Compared to traditional and majoritarian content analysis of written expressions of grief, this approach is original and contributes to provide new literature to a yet scarcely explored complementary field (iconography of death and mourning processes) of research. It is performed from the expertise of a research team that is engaged in intensive analysis of social expression of grief, and that has strong implications in the current scenario of the pandemic where internet resources are the 'only' possible scenarios for many grievers.
Introduction provides historical perspective of the current work, and the authors present ethical issues and limitations of the study (i.e. convenience sample; not full representation of a country, etc) in an appropriate manner. Tables present detailed information of the qualitative research and the five thematic areas in a way that it is easy-to-follow for those not used to the thematic analysis, which also is instructive for those reading the article. They illustrate some themes with quotes that provide a sense of realism to the conceptual frames. The work also opens future research and helps to focus it to religiosity/spirituality/believes, coping strategies etc.
Lines 15.6. The data on Italian sample is truncated.
Line 78. (31,32) should be in brackets.
Line 78. (31,32). to created and Overall, grief
Wouldn't be appropriate to include 'grief iconography' in the title? i.e. between Italian and USA grief iconography or so..
Author Response

(The authors gave the same response as above.)

Reviewer 3 Report
Some light revisions of spelling and grammar are necessary throughout the text so as not to detract from this valuable, otherwise well written work. For example, the first line of the introduction should read either "Expressions of online grief are" or "Expression of online grief is".
Author Response

(The authors gave the same response as above.)

Reviewer 4 Report
The manuscript “Grief iconography between Italians and Americans: A comparative study on how mourning is visually expressed on social media” deals with a relevant and increasingly important topic. Focusing mainly on the visual side of grief in social media and providing an international comparison between the US and Italy, it adds to existing research on this topic and makes an interesting contribution to the field of study. I think that the paper in its current form is mostly fit for publication overall, yet there are some points that need revision before the manuscript can be accepted:
- The authors write of a “two-way communication” being typical to new media on page 1 and 2. Since they mainly refer to (semi-)publicly displayed grief posts in the article, it is usually more than a two-way communication that is taking place, with several users participating in the communication. I’m sure the authors are referring to the interactive nature of social media, however, I would recommend adjusting the phrasing and either write of a multi-way communication or using the interactivity term instead.
- In general, the theory section is well-argued and brings together important literature on the topic. The rationale for the international comparison, however, is rather short and appears rather sudden. Since the authors talk a lot about how social media have transformed mourning in the course of digitalization, I think they should in any case – if only briefly – refer to Brubaker et al. (2013) who have described in detail how social media have expanded mourning both temporally, spatially, and socially. Moreover, Wagner (2018) has extended on this differentiation adding a cultural expansion to these trends, which could also help the authors with their argument on the international comparison.
- The empirical study and the findings are informative and interesting to read. I, personally, have trouble with the quantification of qualitative data, which is particularly problematic in my view, since the US sample and the Italian sample differ significantly in number of participants. Also, the displayed percentages repeatedly suggest a representativeness, which is obviously not the case with this sample. The authors should at least reflect on the benefits of this semi-quantifying approach (in contrast to a pure qualitative approach) in the Discussion section to address this issue.
- It seems that there are quite a lot of blanks missing throughout the text, e.g. “Surveyswere sent” “posts.A strong” in the abstract but also continuously in the article. Please proofread the article carefully and look out for missing blanks in particular.
Literature:
Brubaker et al. (2013). Beyond the Grave: Facebook as a Site for the Expansion of Death and Mourning. The Information Society, 29(3), 152–163. https://doi.org/10.1080/01972243.2013.777300
Wagner, A. J. M. (2018). Do not Click “Like” When Somebody has Died: The Role of Norms for Mourning Practices in Social Media. Social Media + Society, 4(1). https://doi.org/10.1177/2056305117744392
Author Response
As you recommended, we have applied your proposed revisions (see the document linked to this form that summarize all the changes).
Thanks for your valuable advice.

Round 2
Reviewer 1 Report
Thank you for considering incorporating my comments into the paper. The current version has clear descriptions of implications and limitations. I would like to recommend to be published in this form.
Author Response
We thank you for your valuable comments and advices.